# Robustly segmenting quadriceps muscles of ultra-endurance athletes with weakly supervised U-Net

**Hoai-Thu Nguyen**[1]                                        hoai-thu.nguyen@creatis.insa-lyon.fr
**Pierre Croisille**[1,2]                                       pierre.croisille@creatis.insa-lyon.fr
**Magalie Viallon**[1,2]                                        magalie.viallon@creatis.insa-lyon.fr
**Sarah Leclerc**[3]                                            sarah.leclerc@creatis.insa-lyon.fr
**Sylvain Grange**[1,2]                                        sylvain.grange@chu-st-etienne.fr
**Rémi Grange**[2]                                                      remgrange1@gmail.com
**Olivier Bernard**[3]                                         olivier.bernard@creatis.insa-lyon.fr
**Thomas Grenier**[3]                                          thomas.grenier@creatis.insa-lyon.fr

[1] *Univ Lyon, UJM Saint-Étienne, INSA Lyon, UCB Lyon 1, CNRS, Inserm, CREATIS UMR 5220, U1206, F-42023, Saint-Étienne, France*

[2] *Department of Radiology, Centre Hospitalier Universitaire de Saint-Étienne, Université de Saint-Étienne, F-42055 Saint-Étienne, France*

[3] *Univ Lyon, INSA Lyon, UCB Lyon 1, UJM Saint-Étienne, CNRS, Inserm, CREATIS UMR 5220, U1206, F-69621, Villeurbanne, France*

## Abstract

In this study, segmentation of quadriceps muscle heads of ultra-endurance athletes was done using a multi-atlas segmentation and corrective leaning framework where the registration based multi-atlas segmentation step was replaced with weakly supervised U-Net. For the case with remarkably different morphology, our method produced improved accuracy, while reduced significantly the computation time.

**Keywords:** unet, weakly supervised, medical image segmentation

## 1. Introduction

Accurate quadriceps muscles segmentation remains challenging due to the lack of clear muscle head boundaries (Prescott et al., 2011), especially in ultra-endurance athletes. We recently have adapted the multi-atlas segmentation with joint label fusion (JLF) and corrective learning (CL) method of Wang and Yushkevich (2013) to segment our dataset (Nguyen et al., 2018) with only 7 atlases. However, the JLF step is extremely expensive in term of computational time and is not robust in the case of large morphological variation among subjects. Here, we proposed to conserve the principal structure of Wang's framework while replacing the multi-atlas segmentation + JLF step with U-Net (Ronneberger et al., 2015).

## 2. Data

Our data included 3D T1-weighted Water (T1W) MR images collected from 50 volunteered athletes from the Tor des Géants Mountain-Ultra-Marathon (MUM) 2014. On 7 T1W volumes, our medical experts delineated the 4 right thigh's quadriceps muscle heads. These 7 images were separated into 3 groups with remarkably different morphologies: 5 young to middle-aged males, 1 elder male and 1 female. The right leg image volumes of the female and

a young male will serve as test set while the remaining will be our training and validation sets. Meanwhile, our test set also includes 3 left thigh images of young male subjects.

**Data augmentation** is done with the help of deformable registration and random B-Spline warping, hence the weakly supervised part (our U-Net will be trained with experts' segmentations and also weak automatically generated ones). To obtain balanced morphologically diverse training data set, the registration was performed using selected athletes not included in the test set.

## 3. Method

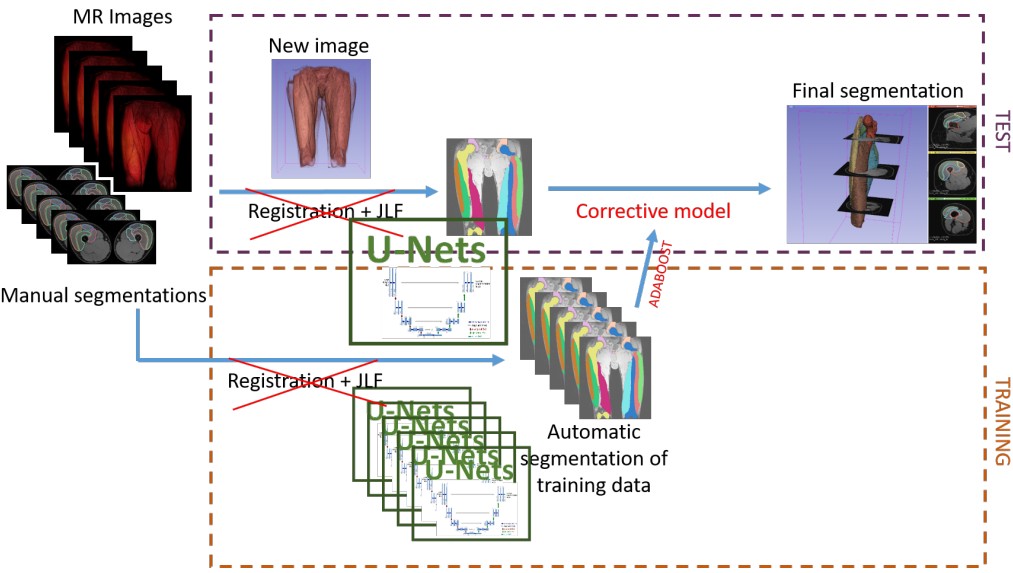

Figure 1: Our segmentation framework based on Wang and Yushkevich (2013)'s

We are conserving the idea behind Wang and Yushkevich (2013)'s algorithm (Figure 1) in which the automatic segmentations of reference images were fed to a CL algorithm that learns and then corrects the typical errors made by the automatic multi-atlas segmentation method. Here, the segmentation method by multi-atlas deformable registration and JLF was replaced with U-Net. The original U-Net of Ronneberger et al. (2015) was used with a Batch Normalization layer after each convolutional layer and ReLU activation function. Before ending with a softmax layer, we added a dropout layer with a coefficient of 0.2 (Li et al., 2018). The axial slices were considered separately and were re-stacked at the end to get the full 3D volume.

We trained and validated a *complete* U-Net with the entire training and validation sets. For the CL step, we trained 5 other U-Nets by using each one of the original right thigh image volumes of the 5 male subjects in the training set for validation. The training set for each of these 5 networks was the original one excluding the image volume used as validation and the volumes derived from it. The segmentation were evaluated using DICE score, Hausdorff distance (HD) and Mean Absolute Distance (MAD) that were computed in 3D for the right thigh volumes and in average of 2D axial slices with available manual segmentations for the left thigh volumes.

## 4. Results

The Figure 2 shows sample results and the Table 4 quantifies results on test set in term of DICE score, HD and MAD. Our approach with U-Net delivered, on one hand, a big improvement in quality of segmentation of the female right leg, on the other hand, a slight decline for male right and left legs. The CL step had, overall, a positive impact on the result, except for the DICE score for the male right leg. This can be explained by the fact that CL is very strong at correcting boundary errors, it is not as efficient in the case of large distanced errors.

Table 1: Evaluation of automatic segmentation approaches: U-Net, U-Net with CL, JLF and JLF with CL.

| Metrics | | Female | Male | Left |
|---|---|---|---|---|
| *DICE* | | | | |
| | U-Net | 0.905 | 0.930 | 0.932 |
| | U-Net + CL | **0.918** | 0.925 | 0.937 |
| | JLF | 0.784 | **0.937** | 0.935 |
| | JLF + CL | 0.829 | **0.937** | **0.938** |
| *HD (mm)* | | | | |
| | U-Net | **22.094** | 44.836 | 18.671 |
| | U-Net + CL | 22.520 | 30.794 | **13.665** |
| | JLF | 35.842 | 18.156 | 16.851 |
| | JLF + CL | 33.568 | **17.694** | 19.707 |
| *MAD (mm)* | | | | |
| | U-Net | 2.726 | 2.407 | 2.422 |
| | U-Net + CL | **1.638** | 1.657 | 1.795 |
| | JLF | 7.921 | **1.039** | 1.975 |
| | JLF + CL | 5.600 | 1.061 | **1.774** |

The U-Net approach have significantly reduced the computation time compared to the one of Wang and Yushkevich (2013). The JLF algorithm took 20 hours to produce a segmentation for one volume with 16 CPUs running in parallel. Training for CL in the case of 5 atlases took 100 hours. Meanwhile, each one of our U-Nets took only 2 hours of training on a Tesla V100 SXM2 32 GB.

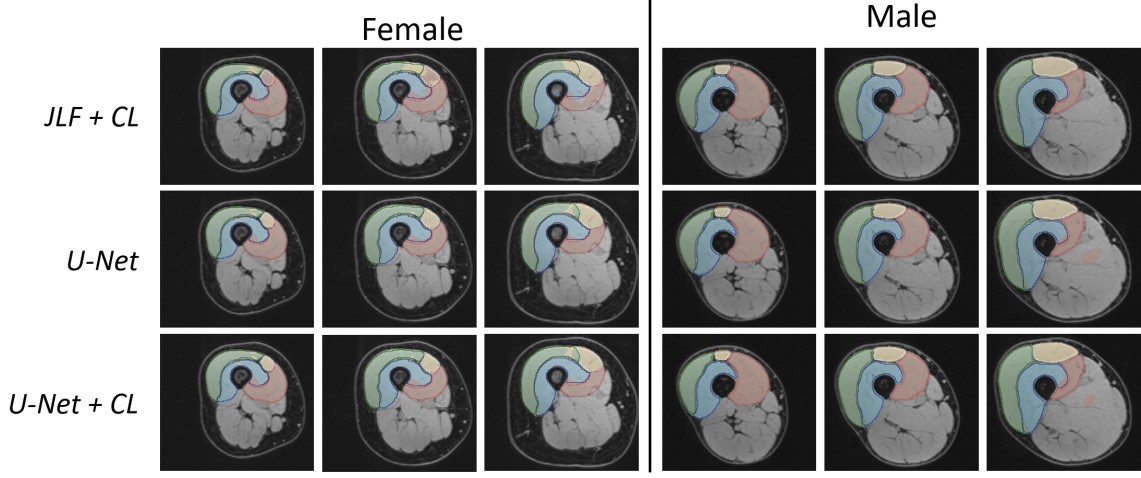

Figure 2: Results of different automatic segmentation approaches (JLF+CL, U-Net and U-Net+CL) on our test subjects. True segmentations are colored contours.

## 5. Conclusion

The proposed segmentation based on U-Net and corrective learning provided similar or even improved accuracy with substantially reduced computation time especially in the case of diverge anatomical morphologies. This automatic segmentation will allow us to investigate the local changes in individual quadriceps heads; hence, the possibility of monitoring the damages in each individual muscle heads along the MUM. An improvement with U-Net 3D and revisited corrective learning is in progress.

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
