# OpenReview forum: "Robustly segmenting quadriceps muscles of ultra-endurance athletes with weakly supervised U-Net"
_MIDL.io/2019/Conference/Abstract — MIDL Abstract 2019_

### Official Review · AnonReviewer2 · 2019-04-27
**Good work to combine multi-atlas segmentation with deep learning**

**Rating:** 3
**Confidence:** 3

**Review:**

It is an interesting piece of work to incorporate traditional multi-atlas segmentation into a deep learning framework.

Cons:
I am not sure why we need several U-nets to simulate the JLF and CL steps. Maybe we could simply concatenate everything to learn segmentation end-to-end.

---

### Official Review · AnonReviewer1 · 2019-05-01
**why not end-to-end with only deep networks?**

**Rating:** 3
**Confidence:** 2

**Review:**


This paper presents an approach for segmentation of quadriceps muscle heads of ultra-endurance athletes. The approach is based on a multi-atlas segmentation framework followed by a corrective leaning framework, where the registration based multi-atlas segmentation step was replaced with weakly supervised U-Net. It may be quite small technical novelty here, but the application is novel, and of a note, the problem is challenging given the extreme variation of muscle morphology. Results are promising, experiments are convincing (despite lack of details caused by the page limitation)

question: why not train the images end-to-end manner without having atlas based segmentation strategy? why not relying on only deep networks with semi-supervised settings directly ?

---

### Decision · Program_Chairs · 2019-05-06
**Acceptance Decision**

Accept